# The acceptability of addressing alcohol consumption as a modifiable risk factor for breast cancer: a mixed method study within breast screening services and symptomatic breast clinics

Julia Sinclair,[1] Mark McCann,[2] Ellena Sheldon,[3] Isabel Gordon,[4] Lyn Brierley-Jones,[5] Ellen Copson[6]

For numbered affiliations see end of article.

**Correspondence to**
Professor Julia Sinclair;
julia.sinclair@soton.ac.uk

## ABSTRACT

**Objectives** Potentially modifiable risk factors account for approximately 23% of breast cancers, with obesity and alcohol being the two greatest. Breast screening and symptomatic clinical attendances provide opportunities ('teachable moments') to link health promotion and breast cancer-prevention advice within established clinical pathways. This study explored knowledge and attitudes towards alcohol as a risk factor for breast cancer, and potential challenges inherent in incorporating advice about alcohol health risks into breast clinics and screening appointments.

**Design** A mixed-method study including a survey on risk factors for breast cancer and understanding of alcohol content. Survey results were explored in a series of five focus groups with women and eight semi-structured interviews with health professionals.

**Setting** Women attending NHS Breast Screening Programme (NHSBSP) mammograms, symptomatic breast clinics and healthcare professionals in those settings.

**Participants** 205 women were recruited (102 NHSBSP attenders and 103 symptomatic breast clinic attenders) and 33 NHS Staff.

**Results** Alcohol was identified as a breast cancer risk factor by 40/205 (19.5%) of attenders and 16/33 (48.5%) of staff. Overall 66.5% of attenders drank alcohol, and 56.6% could not estimate correctly the alcohol content of any of four commonly consumed alcoholic drinks. All women agreed that including a prevention-focussed intervention would not reduce the likelihood of their attendance at screening mammograms or breast clinics. Qualitative data highlighted concerns in both women and staff of how to talk about alcohol and risk factors for breast cancer in a non-stigmatising way, as well as ambivalence from specialist staff as to their role in health promotion.

**Conclusions** Levels of alcohol health literacy and numeracy were low. Adding prevention interventions to screening and/or symptomatic clinics appears acceptable to attendees, highlighting the potential for using these opportunities as 'teachable moments'. However, there are substantial cultural and systemic challenges to overcome if this is to be implemented successfully.

### Strengths and limitations of this study

► A mixed methodological approach enabled a broader exploration of knowledge and attitudes to alcohol as a risk factor for breast cancer than achievable using a single research method.

► Participants were women attending the NHS Breast Screening Programme (NHSBSP) mammograms, symptomatic breast clinics and healthcare professionals based in a single hospital, to understand how attitudes and information needs in this area may differ across these groups.

► Cross-sectional survey results were explored in a series of five focus groups with women and eight semi-structured interviews with health professionals, to enable triangulation of data.

► Research in this area is at an early stage, such that there are no standardised tools to measure the constructs explored.

► The study was conducted in a single UK centre, limiting the potential generalisability of the results.

## BACKGROUND

Breast cancer is the most common cancer in the UK with >54 000 new cases diagnosed annually and the incidence is continuing to rise.[1] Although cure rates are now high, breast cancer remains a significant public health problem and more than 11 000 women per year die of advanced disease.[1] There is an urgent need to focus on developing population-based prevention strategies to reduce the morbidity and mortality from breast cancer.[2]

Alcohol use is now estimated to be responsible for between 5% to 11% of breast cancer cases and current evidence suggests that it is a risk factor for all age groups.[3 4] Alcohol increases the risk of breast cancer in a dose-dependant fashion from low levels of consumption[5 6] and a positive association between alcohol consumption and breast cancer risk

**BMJ**

has been supported by over 100 studies.[7] Over 20% of women aged 45 to 64 reportedly drink more than 14 units per week,[8] so any interventions to reduce population level consumption could have a significant influence on breast cancer rates,[9] as well as help to manage the side-effects of treatment, and improve the overall health of survivors.[10 11]

Two important components for motivating and sustaining behaviour change are the awareness of the health risks (or benefits) of a behaviour, and one's own risk level.[12 13] Awareness of alcohol as a risk factor for breast cancer is low,[14] as is the ability to estimate the alcohol content of commonly consumed drinks and therefore objectively assess one's own alcohol consumption level.[15] This suggests that many women may be unaware that their level of alcohol consumption may be increasing their risk of breast cancer.

Evidence for lifestyle advice interventions in breast cancer prevention is limited,[16] but out-patient clinics for investigation of breast symptoms could constitute what has been called a 'teachable moment'; these are defined as 'naturally occurring health events thought to motivate individuals to spontaneously adopt risk-reducing health behaviours'.[17] Initial research suggests this has high levels of acceptability with women coming for breast mammography screening[17] and provide an opportunity for delivering low-intensity interventions to a general population cohort, as already demonstrated in emergency departments and other healthcare settings.[12 13]

The NHS Breast Screening Programme (NHSBSP) provides free breast mammographic screening every 3 years for all women aged 50 and over. In total, 1.94 million women aged 45 and over were screened within the programme in England in 2011 to 2012 with a 73.1% uptake of routine invitations.[18] In addition, approximately 230 000 women attend a symptomatic breast clinic each year in the UK.[19] Early detection of cancers occurs in very small proportions (0.08% in the screening group, <8% in the symptomatic group are diagnosed with a breast cancer), therefore the vast majority of women who attend do not currently benefit individually from their attendance in prevention terms. This interaction provides an opportunity to reach a significant number of women, in a situational context where the motivation for behaviour change is enhanced, thereby improving the cost-benefit ratio of the current service.

Despite the positive potential of implementing interventions in this setting, it is important to first understand how features of the breast health appointment context may influence the applicability - and acceptability to patients and staff - of any intervention.[20] This study aimed to assess the acceptability, to female patients, and healthcare staff, of offering an alcohol brief intervention (ABI) in NHS clinical breast services. Our research questions were: (a) What level of knowledge do women and staff have about alcohol as a modifiable breast cancer risk? (b) What level of awareness of alcohol content in drinks do women and staff possess? (c) What are women's perceptions and emotions in relation to alcohol and cancer risk? and (d)

How acceptable would an alcohol brief intervention be to staff and women attending breast health appointments? These data will be used to inform the sample size for a definitive project as currently there are no published data on variability or effect sizes in relation to alcohol breast cancer awareness or risk factor knowledge.

## METHODS
### Design
This mixed-method study used quantitative data as the primary frame, with a secondary qualitative frame to build on the quantitative data. Outcomes of the analysis of cross-sectional survey data were used as prompts for focus group and telephone discussions.

Recruitment based on securing a minimum sample size of 100 for each clinical group and 20 staff members was determined to be plausible based on known attendance and staff numbers at Southampton breast services, and to be sufficient to generate data of risk factor and alcohol knowledge to be explored in depth within qualitative interviews.[21]

Staff interview questions about risk factors mirrored the survey questions and additionally explored meaning and emotional appraisal. Three focus groups consisting of a maximum of eight women each were planned, as eight is considered the optimum focus group number when the topic is of importance to participants.[22] Women self-selected and attended the group most convenient for them.

The study received approval from the NRES Committee South Central - Hampshire A Research Ethics Committee (reference no.14/SC/1399), and all participants gave informed consent.

### Patient and public involvement
The design for this study was developed at an innovation workshop funded jointly by Cancer Research UK (CRUK) and BUPA Foundation (a private healthcare charitable trust) which included patient and public involvement (PPI) representation. Following up the initial survey results with participants in a series of focus groups and telephone interviews ensured participants were actively involved in the conduct of the study. The data and preliminary findings were presented at a CRUK showcase event including a wide range of PPI stakeholders.

### Participants
Participants were recruited from three different populations: (1) Women attending NHSBSP mammograms (SG), (2) women attending symptomatic breast clinics (CG) and (3) NHS health professionals associated with these services (HP). All participants meeting the eligibility criteria for inclusion (over the age of 18) on the 12 days the researcher (ES) was in the specified breast health services were approached, and those with sufficient English to give informed consent were recruited. Uptake rate was 82% in SG, 88% in CG and 71% in HP groups, suggesting limited bias in sampling. In the event five

focus groups were conducted with a total of 29 women. The smallest group size consisted of three women and the largest 11. All data were collected between January - July 2015. All patients provided written informed consent before completing the survey.

## Quantitative data

As there are currently no psychometrically validated tools to measure knowledge of breast cancer risk factors or estimation of alcohol content in UK drinks, we developed our own questionnaire specifically for this study (see online supplementary appendix A). Participant demographic information was collected using standardised UK national statistics demographic categorisation. Based on the authors' knowledge of the area and a scoping review of the literature, current known modifiable and non-modifiable risk factors for breast cancer were identified and cross-referenced against risk factors reported on public information websites.[23–25] Factors were classified as having convincing, probable, limited or no evidence. This provided a coding framework for survey responses (see online supplementary appendix B). Each free-text response listed by study participants in response to the *question 'write down anything you think might increase the risk of breast cancer'*, was coded by ES, and 10% of the responses were double-coded by JS to check consistency.

Self-reported height and weight were used to calculate body mass index (BMI). Participants were given four multiple choice questions to identify the units of alcohol in four drink types (see online supplementary file A).

The phrase 'We are interested to know your thoughts on the impact of adding specific cancer prevention information to the breast cancer screening process.' was used specifically rather than referring to alcohol education as we did not wish to influence participant's response to other questions within the survey by highlighting alcohol (or obesity) as the two greatest potentially modifiable risk factors for breast cancer. Participants in each of the three groups completed the same survey to allow for comparison across groups

## Qualitative data

Women attending either a clinic or a mammography screening appointment who had taken part in the survey were invited (with the option to bring a female friend or relative) to a focus group '*to discuss their opinion about some of the methods available to try and reduce the number of women who develop breast cancer'*. The topic guides are available in online supplementary appendix C.

Health professionals, working in breast clinics or screening services, were also invited to take part in a semi-structured telephone interview to discuss some of the findings of the survey further. Interviews were by telephone to reduce participant burden and maximise the degree of anonymity for participants to encourage frank and open responses.

## Analysis

Clinic/mammogram attenders (CG and SG groups) reported their views of what may be acceptable and appropriate in terms of information about alcohol as a risk factor for breast cancer, delivered as an intervention embedded within NHS breast clinics (both symptomatic and preventative screening appointments).

After the clinic/mammogram attenders (CG and SG groups) and staff (HP group) data sets were analysed separately, they were compared with each other to determine where the staff and attendee views met and diverged, giving two sides of the same story.

Chi square tests assessed differences between the recruited groups in terms of demographical and health characteristics, risk factor identification and ability to identify alcohol units. Likelihood ratio tests comparing logistic regression models with and without interaction terms were used to assess if the association between risk factor identification and individual characteristics varied between groups. Analysis were conducted using Stata V.14.2.

Qualitative data from the focus groups and telephone interviews were transcribed verbatim, and then analysed using the principles of thematic analysis[26] whereby themes emerging from the data are synthesised into initial categories. These initial categories were then discussed with the wider research team for identification of the most salient themes with reference to the overarching research question and linkage back to the quantitative survey data. During data collection and analysis there were discussions among the research team to increase reliability of decisions made regarding categories and interpretations. The data were then revisited so that as many initial categories as possible could be synthesised into secondary themes. The analysis presented and quotations included are to give a flavour of the data, not to 'prove' the analysis.

## RESULTS
### Survey findings

In total 238 people completed the survey; 102 NHSBSP attenders (SG), 103 symptomatic breast clinic attenders (CG) and 33 breast unit staff (HP). Other than the differences that would be anticipated between the groups due to the populations they were recruited from (age, employment status and breast treatment histories) smoking was significantly less frequent in the staff compared with the attender group (0% vs 10% current smokers, 88% vs 53% never smokers, p=0.012). An overview of the characteristics of the participants in the three samples is shown in table 1.

### Screening (SG) and symptomatic clinic (CG) samples

There were significantly more women under 50 and over 70 years in the symptomatic clinic group than attended screening. In terms of knowledge of risk factors, there was little difference between the two groups (see table 2). Each group identified a mean of 1.8 risk factors (ranging

**Table 1** Demographics of survey participants

| | NHSBSP screening attenders | Symptomatic clinic attenders | Staff | Total |
|---|---|---|---|---|
| | N (col %) | N (col %) | N (col %) | N (col %) |
| **Age group** | | | | |
| 40 or under | 1 (1.0) | 16 (15.5) | 8 (24.2) | 25 (10.5) |
| 41 to 50 | 18 (17.6) | 27 (26.2) | 11 (33.3) | 56 (23.5) |
| 51 to 60 | 50 (49.0) | 23 (22.3) | 13 (39.4) | 86 (36.1) |
| 61 to 70 | 22 (21.6) | 17 (16.5) | 1 (3.0) | 40 (16.8) |
| 71 and over | 11 (10.8) | 20 (19.4) | 0 (0.0) | 31 (13.0) |
| **Relationship status** | | | | |
| Never married | 12 (11.8) | 13 (12.6) | 7 (21.9) | 32 (13.5) |
| Married | 65 (63.7) | 61 (59.2) | 20 (62.5) | 146 (61.6) |
| Separated | 18 (17.6) | 15 (14.6) | 4 (12.5) | 37 (15.6) |
| Widowed | 7 (6.9) | 14 (13.6) | 1 (3.1) | 22 (9.3) |
| **Employment status** | | | | |
| Working | 60 (60.6) | 49 (53.3) | 33 (100.0) | 142 (63.4) |
| Retired | 24 (24.2) | 31 (33.7) | 0 (0.0) | 55 (24.6) |
| Looking after the home or family | 10 (10.1) | 8 (8.7) | 0 (0.0) | 18 (8.0) |
| Long-term sick or disabled | 5 (5.1) | 4 (4.3) | 0 (0.0) | 9 (4.0) |
| **General health** | | | | |
| Good | 79 (79.8) | 71 (78.9) | 30 (90.9) | 180 (81.1) |
| Fair | 16 (16.2) | 15 (16.7) | 3 (9.1) | 34 (15.3) |
| Poor | 4 (4.0) | 4 (4.4) | 0 (0.0) | 8 (3.6) |
| **Treatment history** | | | | |
| Attended screening | 53 (67.9) | 23 (31.9) | 8 (66.7) | 84 (51.9) |
| Investigated for symptoms | 23 (29.5) | 24 (33.3) | 4 (33.3) | 51 (31.5) |
| Treated for breast cancer | 2 (2.6) | 25 (34.7) | 0 (0.0) | 27 (16.7) |
| **Body mass index group** | | | | |
| Healthy/underweight *(<25 kg/m2)* | 32 (35.6) | 35 (45.5) | 15 (50.0) | 82 (41.6) |
| Overweight | 37 (41.1) | 19 (24.7) | 9 (30.0) | 65 (33.0) |
| Obese *(≥30 mg/m2)* | 21 (23.3) | 23 (29.9) | 6 (20.0) | 50 (25.4) |
| **Smoking status** | | | | |
| Current smoker | 11 (10.9) | 9 (9.8) | 0 (0.0) | 20 (8.8) |
| Ex-smoker | 32 (31.7) | 33 (35.9) | 4 (12.1) | 69 (30.5) |
| Never smoked | 58 (57.4) | 50 (54.3) | 29 (87.9) | 137 (60.6) |
| **Alcohol status** | | | | |
| Has drunk alcohol in last 12/12 | 61 (60.4) | 66 (73.3) | 25 (75.8) | 152 (67.9) |
| Has not drunk alcohol in last 12/12 | 40 (39.6) | 24 (26.7) | 8 (24.4) | 72 (32.1) |
| Total | **102** | **103** | **33** | **238** |

NHSBSP, NHS Breast Screening Programme.

from 0 to 8). Obesity was identified as a risk factor by approximately 30% of participants, and smoking by almost 50% in each group whereas alcohol was listed by 16% in the screening group, 23% in the symptomatic clinic group.

**Alcohol**

Personal alcohol consumption was reported by 60% of participants in the screening group and 73% in the clinic group. In those who did not drink alcohol, there was no difference between screening and clinic groups in

**Table 2** Identification of risk factors (RF) for breast cancer in survey participants

| | NHSBSP screening attenders | Symptomatic clinic attenders | Total | |
|---|---|---|---|---|
| | N (col %) | N (col %) | N (col %) | Statistics |
| Identified at least one RF | | | | Pearson $\chi^2$ (1) 0.0164 p=0.898 |
| No | 23 (22.5) | 24 (23.3) | 47 (22.9) | |
| Yes | 79 (77.5) | 79 (76.7) | 158 (77.1) | |
| Modifiable RF identified | | | | Pearson $\chi^2$ (1) 0.0391 p=0.843 |
| Yes | 65 (63.7) | 67 (65.0) | 132 (64.4) | |
| No | 37 (36.3) | 36 (35.0) | 73 (35.6) | |
| Non-modifiable RF identified | | | | Pearson $\chi^2$ (1) 0.8197 p=0.365 |
| Yes | 51 (50.0) | 45 (43.7) | 96 (46.8) | |
| No | 51 (50.0) | 58 (56.3) | 109 (53.2) | |
| Obesity identified as RF | | | | Pearson $\chi^2$ (1) 0.0647 p=0.799 |
| Yes | 31 (30.4) | 33 (32.0) | 64 (31.2) | |
| No | 71 (69.6) | 70 (68.0) | 141 (68.8) | |
| Alcohol identified as RF | | | | Pearson $\chi^2$ (1) 1.8921 p=0.169 |
| Yes | 16 (15.7) | 24 (23.3) | 40 (19.5) | |
| No | 86 (84.3) | 79 (76.7) | 165 (80.5) | |
| Lifestyle identified as RF | | | | Pearson $\chi^2$ (1) 0.3280 p=0.567 |
| Yes | 8 (7.8) | 6 (5.8) | 14 (6.8) | |
| No | 94 (92.2) | 97 (94.2) | 191 (93.2) | |
| Hormone medications identified as RF | | | | Pearson $\chi^2$ (1) 0.3282 p=0.567 |
| Yes | 13 (12.7) | 16 (15.5) | 29 (14.1) | |
| No | 89 (87.3) | 87 (84.5) | 176 (85.9) | |
| Smoking identified as RF | | | | Pearson $\chi^2$ (1) 0.1194 p=0.730 |
| Yes | 49 (48.0) | 47 (45.6) | 96 (46.8) | |
| No | 53 (52.0) | 56 (54.4) | 109 (53.2) | |
| **Total** | **102 (100.0)** | **103 (100.0)** | **205 (100.0)** | |

RF, risk factor; NHSBSP, NHS Breast Screening Programme.

identifying alcohol as a risk factor for breast cancer (15% vs 16%). However, among those who drank alcohol, identifying this as a risk factor for breast cancer was significantly more likely in the symptomatic group (35%) than those attending screening (4%) ($X^2$ 12.7, p=0.0004). See figure 1

Only 88/152 (57.9%) participants who drank alcohol stated that they knew how to estimate the alcohol content in drinks, and of these 72%, 55%, 24% and 24% correctly identified the number of units in a standard glass of wine, pint of beer, litre of cider and bottle of spirits respectively, out of a choice of four options for each one. Overall in both SG and CG there was a significant association between identifying alcohol as a risk factor for breast cancer and personal alcohol consumption (25.2% in those who drink alcohol vs 10.9% in non-drinkers, p=0.031), as well as a positive association with being able to correctly identify the alcohol content of drinks. Of those who got none of the four drink units correct, 86.2% also did not identify alcohol as a risk factor compared with 13.8% who did identify alcohol as a risk factor (p=0.01), suggesting that increased awareness about alcohol is associated with the knowledge necessary (if not sufficient) to facilitate behaviour change (see table 3).

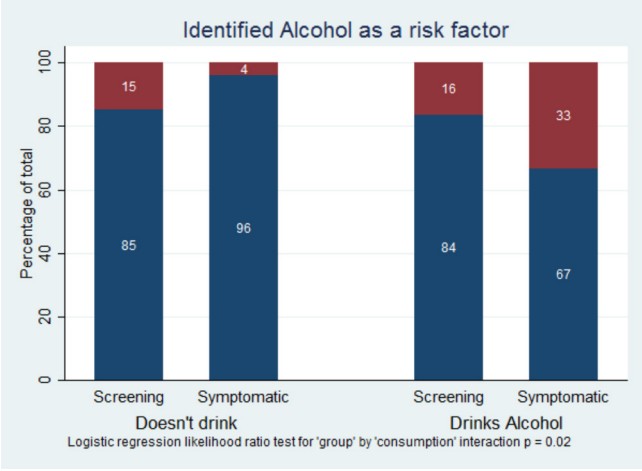

**Figure 1** Identification of alcohol as a risk factor for breast cancer by drinking status in screening and clinic attenders.

**Table 3** Identification of alcohol as risk factor by own alcohol status and knowledge in survey participants

| | Identified alcohol as a risk factor | | | |
|---|---|---|---|---|
| | **No** | **Yes** | **Total** | |
| | **N (row %)** | **N (row %)** | **N (row %)** | **Statistics** |
| Has drunk alcohol in last 12 months | 95 (74.8) | 32 (25.2) | 127 (100.0) | Pearson $\chi^2$ (2) 6.9734 p=0.031 |
| Has not drunk alcohol in last 12 months | 57 (89.1) | 7 (10.9) | 64 (100.0) | |
| Missing | 13 (92.9) | 1 (7.1) | 14 (100.0) | |
| Knows how to find alcohol content (if drinks) | | | | Pearson $\chi^2$ (2) 10.2683 p=0.006 |
| Yes | 57 (70.4) | 24 (29.6) | 81 (100.0) | |
| No | 40 (81.6) | 9 (18.4) | 49 (100.0) | |
| Missing/does not drink | 68 (90.7) | 7 (9.3) | 75 (100.0) | |
| Able to give units in any of four drinks | | | | Pearson $\chi^2$ (2) 9.1821 p=0.01 |
| None correct | 100 (86.2) | 16 (13.8) | 116 (100.0) | |
| One or two/four correct | 52 (77.6) | 15 (22.4) | 67 (100.0) | |
| Three or four/four correct | 13 (60.0) | 9 (40.9) | 22 (100.0) | |
| Total | **165** | **40** | **205** | |

## Obesity

In terms of BMI, based on self-reported height and weight, (see table 1) a minority of each group (36% in SG and 46% in CG group) could be classified within the healthy/underweight range, while 23% of SG (and 30% of the CG) met the criteria for being obese.[27] Approximately one-third of each group identified obesity as a risk factor for breast cancer, however women whose BMI placed them in the obese category, were significantly less likely to list obesity as a risk factor in the unadjusted logistic regression model (OR = 0.365, CI 0.16 to 0.834, p<0.05) with no difference between screening and symptomatic groups.

## Potential impact of adding prevention information

The response to adding a '5 min cancer prevention information session' to either the SG or CG was similar in both groups 60/197 (30.5%) said it would make them more likely to attend and 137/197 (69.5%) said it would make no difference to their attendance. No one marked the option that it would make them less likely to attend. Potential disadvantages of adding a cancer prevention session were identified by 28% of the SG and 40% of CG, with time, additional use of resources and potential cause for anxiety cited as the main concerns. In terms of preference for how this might be delivered, 31% stated they would prefer this to be by an electronic device, 26% by post, 40% by a trained nurse and 18% by a trained volunteer.

## Health professional sample

Thirty-three health professionals (HPs) participated and key findings are summarised in table 1. Seventy-three per cent of the HP sample named at least one risk factor for developing breast cancer. Obesity was identified by 19/33 (58%) and alcohol consumption was identified by 17/33 (52%)HPs. 45% (15/33) stated they knew how

much alcohol was in a drink, and correct answers (out of a choice of four) were given by 21% to 61% of the sample for four commonly consumed drinks. Similarly to the SG and CG samples the majority of staff (22/33- 67%) felt that adding a cancer prevention session would make no difference to attendance, but less (5/33- 15%) thought it would encourage attendance and one HP thought it would discourage people from attending. A far higher percentage of the HP group than the CG/SG participants listed potential disadvantages of adding a cancer prevention session (82%). In addition to time, use of additional resources and potential cause for anxiety the staff group also mentioned contributing to 'the worried well' culture, time inefficiencies and fears about it being seen as 'blaming' or patronising' as potential disadvantages. In terms of preferred method of delivery 52% stated they would prefer this to be by an electronic device, 30% by post, 18% by a trained nurse and 9% by a trained volunteer.

## Qualitative findings

In accordance with the open ended focus group questions (see online supplementary appendix C) themed around the potential introduction of cancer risk messages to symptomatic clinics, what follows are the principle emergent themes from the focus group discussions and interviews.

## Trust, knowledge and uncertainty in relation to cancer risk factors

In line with the results of the main survey women showed little knowledge of alcohol as a risk for breast cancer, they also demonstrated considerable uncertainty about which sources of information they could trust. Women's knowledge of breast cancer tended to be based on their own experiences or stories told by others they knew, which in turn could encourage them to find out more. In focus

groups they exchanged stories and talked of what they had learnt; they appeared to trust this type of first-hand information and started engaging with it by asking questions.

The conversations suggested a real lack of knowledge about how managing alcohol consumption fitted into a healthy lifestyle, or even how to talk about it: how much was safe; whether it was a risk at all; how many units were in a measure and understanding different sized drinks and strengths of alcohol.

> I mean a glass of wine's… it's a bit like a cup of tea… a glass of wine used to be like that, **[hand movement demonstrates small size]** now it's like that (**[hand movement demonstrates bigger size]** when you ask just for a glass they vary enormously (FG4, p1).

Both women and staff recognised a need to understand more about lower risk alcohol consumption specifically relative to volumes of drinks consumed. Women's motivations, understanding and rationalisation for their own alcohol consumption were varied and complex and perhaps reflect the varied cultural values, symbolic value and messages about alcohol represented in today's social environment.

Women's expectations of the role of health professionals in providing health education were also apparent here; they said they were happy for staff to advise them in breast screening clinics, hinting at a certain passivity in their own role in seeking/receiving alcohol information of this kind.

> I think having any kind of information about it. I mean I haven't heard anything about breast cancer at all. Not from my GP – no one's mentioned it once. So I think it's just having any kind of information at all would be a good place to start (FG4, p.8).

Both women and staff wanted consistent, evidence-based facts about alcohol consumption, and specific risks related to breast, rather than other forms of cancer

> …People aren't aware exactly how much they drink and also are surprised when they find out that alcohol is a risk factor for breast cancer (iv7, p3).

Participants felt there were many mixed messages about the risks of alcohol consumption, and wanted clear information from a trusted source

> it's been marketed in such a way that, particularly with wine, that it's good for you, in a way (iv5, p1).

Both staff and women saw concrete, evidence-based information as a necessity in a field where there is an overwhelming amount of uncertain information, which can cause anxiety.

However, despite appearing to value and engage with information gained from peers in focus groups, women talked of finding it hard to know how to view conflicting information from media, hearsay and their own experiences. Women who talked of knowledge gained from lived experiences

> I had it (breast cancer) 10 years ago. I don't drink, I don't smoke, and I still got it (FG4, p6)

had views that were seemingly less flexible and demonstrated a lack of trust in 'evidence based' preventative information.

Staff echoed this uncertainty, highlighting their own and women's confusion; they reported needing evidence about alcohol consumption and its specific links to breast cancer to feel more confident advising patients. Staff also displayed their own relatively low levels of literacy around alcohol in relation to breast cancer when talking about what they felt the content of preventative information should be.

> …the whole idea of trying to prevent cancer through lifestyle seems like it's quite in its early stages…it's not just the public that we need to spread the word to but it's the healthcare professionals too because people are not really aware of it (iv2, p3).

Health professionals were fearful of offering false advice, again underlining their mixed views about the legitimacy of including preventative information in their role as health professionals and a strong concern

> you just have to be careful you don't put women off coming to screening… (iv8, p2),

yet women did not mention this as an issue, confirming the survey results presented above; rather that they wanted more opportunities to access information.

Staff also underlined their own need for training about levels of alcohol consumption and its bearing on breast cancer to improve their confidence delivering advice to patients.

Lack of time to engage with information was also a factor for women - they highlighted the need for succinct, universally appealing information that would engage across different groups in society. In a similar vein to the pink ribbon used for fundraising by breast cancer charities, staff suggested the idea of a symbol to use to catch women's attention for prevention information.

### The need for cultural change and individualised health messages

Both staff and women recognised a need for a wider cultural change before fully accepting the need to accept, act on and acknowledge alcohol as a modifiable risk for breast cancer.

> I think as a nation we are not very honest about the harms of alcohol, and that goes across the board in terms of doctors and nurses as well (iv3, p2).

There appeared to be an unacknowledged 'collusion of denial' where staff and women avoided being proactive in discussing alcohol as part of the dialogue around preventing breast cancer

> I mean when is the best time for health promotion, I don't know (iv4, p1).

This suggests a challenge of dealing with alcohol as a risk for breast cancer on both sides, where women seemed to assume staff would tell them about risks that are important and where staff would not ask women about their alcohol intake as they did not see it as part of their role.

Staff agreed that breast screening appointments could be ideal opportunities for preventative information

because you've already got yourself a self-selecting audience, people who feel it's important enough to turn up(iv4, p2),

though did not see it as part of their role to deliver this information.

A key motivator for women appears to be whether a risk factor applies to them specifically

If you don't own it, you're not going to act on it are you? (FG3, p25).

In order to make informed choices about how to prevent breast cancer they reported wanting personalised information in order to address the risks they knew about, and if it was meaningful to them, they seemed more motivated to make the changes. Confusion between relative risks of individual risk factors, population based risk factors and definite 'causes' were common.

Any rationale given for changes specifically to prevent breast cancer was primarily from women with lived experience of breast cancer (themselves or a close family member). They had either been advised by health professionals to make changes, or made aware by their affected family member:

I've got it on both sides of my family, …I…check what I eat….and I don't drink a lot anyway (FG2);

they seldom talked of changes to alcohol consumption.

Overall, women with a family history of breast cancer reported being less inclined to make lifestyle changes, especially if relatives had adopted healthy lifestyles and had still got breast cancer, re-enforcing a more fatalistic view

my GP's told me of various risk factors but then it happened to my mum anyway. So I shrug my shoulders (FG4, p5) and there's certain lifestyle things I can do but then I have, like, this little thing on my shoulder going 'well it didn't make any difference for my mum (FG5, p4).

Women suggested the use of humour as a way of making breast cancer prevention information more engaging and accessible

…There should be more discussion between mothers and daughters … like fathers and sons. . 'have you checked your balls son?' [laughter] There should be a light-heartedness about it rather than it being you know when you see these adverts and it's all tears … (FG4, p33),

although this was in the context of self-examination rather than addressing alcohol consumption.

## Tensions in discussing alcohol consumption as part of breast cancer prevention

In sharp contrast to the discussions of women in the focus groups, a common theme that ran through the interviews with staff was an ambivalence about discussing alcohol use with women who came to screening or symptomatic breast appointments. This ambivalence appeared to reflect a number of underlying processes. These are described here in preliminary form, but will require much greater exploration in future research (See figure 2).

### Perceptions of responsibility

Staff recognised that prevention and health promotion was important, but did not see it as their role. Alcohol use was described as

the last thing (staff) would discuss with patients (iv1).

There was a certain sensitivity and worry about raising the topic.

People don't like to think they're being lectured to, patronised… it's quite a careful balance…, between imparting useful information in a way that people are going to be receptive to or people feeling that you're sort of 'nanny stating' them (iv8, p2).

There were also concerns about giving advice based on population risk and the inherent uncertainties between cause and effect that this contains

they might do (what is suggested)… they think they've done everything right and then they still get breast cancer (iv2, p2).

Staff agreed with the idea of health promotion, in principle

…it is important but maybe not essential"(iv8, p1),

but often felt the responsibility did not lie with them (as also shown in the survey results). This seemed connected with seeing themselves responsible for diagnosis and treatment rather than prevention. There was ambivalence as to whether it was their role to encourage patients to take a preventative approach, by advising about lifestyle risk factors

I don't really think it's our role to do that(iv7, p3).

This tension between reactive and preventative health roles may be a barrier to integrating health prevention within already established screening and treatment services as part of a cancer prevention culture.

We're very good at just focusing on the disease and not telling the patient what they can do to help (iv6, p1).

There was also a suggestion that different competencies may be required to fulfil the role:

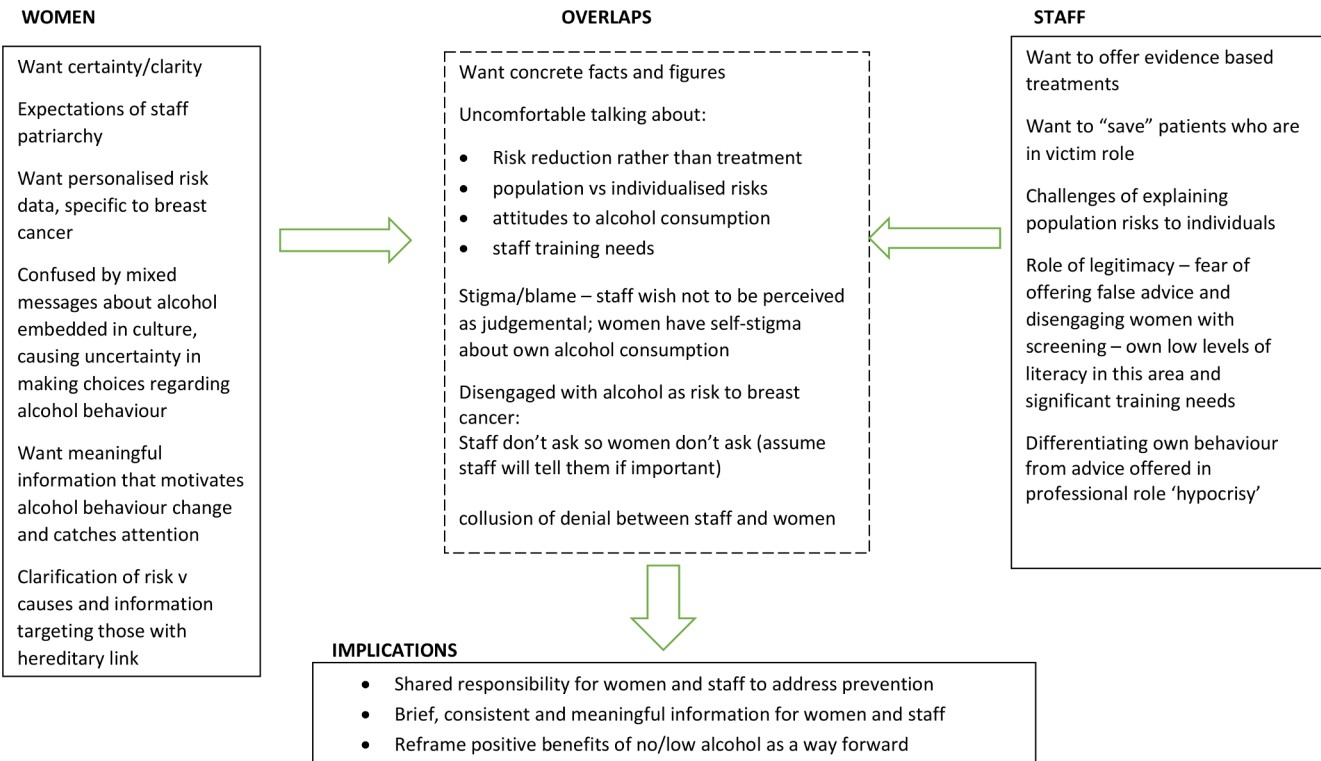

**WOMEN**

Want certainty/clarity

Expectations of staff patriarchy

Want personalised risk data, specific to breast cancer

Confused by mixed messages about alcohol embedded in culture, causing uncertainty in making choices regarding alcohol behaviour

Want meaningful information that motivates alcohol behaviour change and catches attention

Clarification of risk v causes and information targeting those with hereditary link

**OVERLAPS**

Want concrete facts and figures

Uncomfortable talking about:

- Risk reduction rather than treatment
- population vs individualised risks
- attitudes to alcohol consumption
- staff training needs

Stigma/blame – staff wish not to be perceived as judgemental; women have self-stigma about own alcohol consumption

Disengaged with alcohol as risk to breast cancer:
Staff don't ask so women don't ask (assume staff will tell them if important)

collusion of denial between staff and women

**STAFF**

Want to offer evidence based treatments

Want to "save" patients who are in victim role

Challenges of explaining population risks to individuals

Role of legitimacy – fear of offering false advice and disengaging women with screening – own low levels of literacy in this area and significant training needs

Differentiating own behaviour from advice offered in professional role 'hypocrisy'

**IMPLICATIONS**

- Shared responsibility for women and staff to address prevention
- Brief, consistent and meaningful information for women and staff
- Reframe positive benefits of no/low alcohol as a way forward
- Culture-wide approach to alcohol consumption necessary, not just NHS

**Figure 2** Ideas map: engaging with alcohol as a risk factor for breast cancer.

for those of us that are trained in symptom control or diagnostic activities, to be a health promoter is a very, very different kettle of fish and a very different training I would imagine (iv4, p1).

Women however did not appear to make this differentiation within the health professional role, and saw them all as a trusted sources of advice.

I definitely think the preventative line you're going down is far better than the reactive system we have at the moment (woman, FG4, p24).

A cultural mind-set underlying alcohol was considered to be a key barrier to acceptance of it as a modifiable risk factor for breast cancer

I can't quite see…, a bottle of Lanson Champagne, saying on the bottle, you know this is a killer, don't - don't drink [laughter] (FG4, p4).

With one exception, there was difficulty in talking about alcohol use except as a significant problem for a minority of people

it would be important to reinforce the idea that a lot of people drink too much that aren't alcoholics and highlight the fact that it's very normal (iv3, p1),

The need for a cultural shift in attitudes to alcohol-related behaviour as well as towards making preventative lifestyle choices was acknowledged in order to make conversations around alcohol and health less problematic.

I think undoubtedly people will bring their own prejudices to wherever they are, and clearly, people don't like to feel hypocritical either do they, so I think the best bet is to try and be honest to people (iv8, p1).

### Stigma of breast cancer and alcohol

Avoidance of risks was one way women reported gaining control over getting cancer, but this also related to feelings of guilt if they saw it as their fault for not avoiding something

You think what might I have done that's caused this? (FG2, p33).

If a risk of cancer was seen as genetic then women tended to feel they were immune from responsibility or blame for not making healthy lifestyle choices. One woman alluded to the judgement that might be made about someone with breast cancer

I just think [of them as] someone who's got it wrong somewhere along the line. But then you've got – you know – genes that - that get passed on from your family, so then I'd be wrong in that instance (FG4, p1).

This provokes questions around women's feelings of guilt and blame for making the 'right' and 'wrong' lifestyle choices and also show how stigma can exist even in groups of women who have experienced cancer themselves.

## DISCUSSION

This study demonstrates the potential acceptability and need for the development of a contextualised ABI as part of encouraging a cancer prevention culture in breast health settings. The qualitative findings illuminate where women's and staff views meet and diverge. This will inform the development of an appropriate intervention that engages women with alcohol information, giving them feedback about their alcohol consumption in a breast health context. This can be used as an opportunity to improve awareness of alcohol as a risk factor for breast cancer, and improve skills in monitoring alcohol consumption.

Although the exact mechanism by which alcohol acts to promote carcinogenesis is unclear, epidemiological evidence that alcohol is a significant and modifiable dose dependent risk factor for breast cancer is now widely accepted by the scientific community. However, our data indicate that this information is not known by the majority of women attending breast screening and symptomatic clinics. These results are similar to others which have examined awareness of alcohol as a risk factor for breast cancer,[14] and also studies demonstrating low levels of knowledge about alcohol content of drinks in the public and health professionals

This is particularly important as recent changes to UK Chief Medical Officers' low risk drinking guidelines[15] aimed to publicise the concept that there is no safe level of drinking and highlighted the potential long-term risks associated with alcohol intake even at low levels, including the development of certain cancers, including breast cancer.

The Health Survey for England (2014) indicates that almost 80% of women had drunk alcohol at some point during the previous year; the percentage of women drinking over 14 (UK) units per week was highest in the 45 to 54 year (20%) and 55 to 64 year (22%) age groups and the proportion of people drinking at increased-risk-levels was greatest in the highest income groups.[8] Home drinking is an embedded social practice, which may be resistant to change,[28] and this normalisation of alcohol use by health professionals may account for some of the ambivalence they have to discuss alcohol consumption as a risk factor for breast cancer with patients. However, it is also important to acknowledge the 'alcohol harm paradox' which shows that although people in lower social classes drink less, they have worse alcohol related morbidity.[29] This is associated with a clustering of other factors in more deprived groups, such as obesity, smoking and poor diet and exercise, which are also relevant modifiable factors for breast cancer prevention, and may have an impact on how they receive and act on health messages.[23] However, the social gradient is reversed in terms of breast cancer incidence,[30 31] suggesting that any intervention targeted at reducing alcohol consumption should have benefits across the female population and will not worsen any health inequalities.

Both staff and clinic attendees showed ambivalence about discussing alcohol, concerned about it being taken as stigmatising or blaming women, demonstrating a wider social ambivalence about our relationship with alcohol. However, this study and the predicting risk of cancer at screening (PROCAS) clinical trial, which is exploring the potential to produce personalised breast cancer risk assessments, have both confirmed that it is feasible for women to interact with a brief and unsupervised health survey while attending routine screening appointments, despite the short appointment times provided in this setting.[2] Most importantly, our data confirm that women would not be put off attending breast screening or clinic appointments if they were aware they would receive some cancer prevention education and many in fact suggesting that they would welcome this intervention with over 30% indicating that this would make them more likely to attend their appointment. Data collected in this study have been used to inform a logic model which will underpin the development of an intervention prototype.

The effectiveness of ABIs to reduce alcohol intake in people drinking at non-dependent levels has been investigated extensively, with good evidence for effectiveness.[32 33] While historically, ABIs have been provided by professionals, there are now increasing numbers of online tools which aim to provide screening, monitoring and support.[34] There is some evidence that women and young people may find this a more attractive option than a face-to-face session.[35 36] There is also evidence that simply answering questions on alcohol intake can result in changing subsequent self-reported behaviour.[37]

In 2008, Demark-Wahnefried *et al* published guidelines encouraging physicians to use 'teachable moments' to provide patients with lifestyle advice aimed at cancer prevention, regardless of the patient's motivation to receive the message.[38] It is interesting to note that while it has become routine practice to assess patients in breast clinics for a possible inherited susceptibility to breast cancer and refer on for family history or genetics investigation, there is currently no equivalent pathway for patients who have potentially significant modifiable lifestyle risk factors, including alcohol use.

### Strengths and limitations

The study is limited in that it was carried out in a single centre and was cross-sectional in nature. Given the lack of research in this area, a survey utilising a range of response modes, including free text responses, was used to explore knowledge of alcohol as a risk factor for breast cancer in the absence of a validated tool. However, a significant strength of this study is the use of mixed methods to explore in greater depth the findings from the survey, to better understand the ambivalence that attendees and staff have to discuss alcohol use as part of cancer prevention.

## CONCLUSION

This study confirms that knowledge of alcohol as a modifiable risk factor for breast cancer is low among women attending symptomatic breast clinics or routine NHSBSP mammograms. A lack of literacy regarding the alcohol content of commonly consumed drinks indicates that many women are not well equipped to assess their own alcohol intake. However, many women participating in this study reacted positively to the suggestion of adding information on cancer prevention, in general, to NHS breast screening or clinic attendances, although there was ambivalence by staff to delivering it. Full utilisation of these ' teachable  moments ' will require education of healthcare professionals, as well as a wider cultural change around alcohol use. Further research is also required to understand how best to embed a prevention culture, which includes giving clear non-judgemental information about the relative risks of alcohol consumption, within current health systems.

Please include Figure legends/captions at the end of your main manuscript

**Author affiliations**
[1] Clinical and Experimental Sciences Academic Unit, Faculty of Medicine, University of Southampton, Southampton, UK
[2] MRC/CSO Social and Public Health Sciences Unit, University of Glasgow, Glasgow, UK
[3] Core Trainee in Psychiatry, Peninsula Postgraduate Medical Education, Plymouth, UK
[4] Department of Pharmacy Health and Well-being, University of Sunderland, Sunderland, UK
[5] Department of Sociology, University of York, Wentworth College, York, UK
[6] Cancer Sciences Academic Unit, Faculty of Medicine, University of Southampton, Southampton, UK

**Acknowledgements**  Many thanks to all of the delegates at the CRUK Innovation workshop for their input and feedback during the conceptualisation phase of the project. For Lucy Rocca and the members of Soberistas, www.soberistas.com, for their contributions to the survey. For the staff and women attending appointments who also gave their time, and to Magda Novak, Peter Dutey-Magni and Jennifer Allen who assisted in the investigation and data curation for the study.

**Contributors**  Based on casrai.org/credit taxonomy: Conceptualisation: JS, EC; Funding acquisition (CRUK innovation workshop): JS, MMcC, LB-J, EC; Methodology: JS, MMcC, LB-J, EC; Supervision: JS, EC; Investigation & Data curation: IG, LB-J, ES; Analysis, Validation & visualisation: MMcC, ES, IG, JS, EC; Writing and editing: All.

**Funding**  The CRUK / BUPA Foundation Fund (Innovation Grant – 2014) grant number 19626 supported this work. MMcC holds a Medical Research Council/University fellowship supported by an MRC partnership grant (MC/PC/13 027) and is part of the MRC/CSO SPHSU Complexity programme (MC_UU_12017/14 / SPHSU14).

**Competing interests**  ES, IG and LB-J have nothing to disclose. MMC reports grants from Medical Research Council, grants from Medical Research Council/Chief Scientist Office, during the conduct of the study. JS reports grants from the Medical Research Council, NIHR during the conduct of this study. EC reports personal fees from World Cancer Research Fund for acting as grant panel member, outside the submitted work.

**Patient consent for publication**  Obtained.

**Ethics approval**  The study received approval from the NRES Committee South Central - Hampshire A Research Ethics Committee (reference no.14/SC/1399), and all participants gave informed consent

**Provenance and peer review**  Not commissioned; externally peer reviewed.

**Data sharing statement**  An anonymised data extract is available at Open Science Framework https://osf.io/fhvpk/DOI:10.17605/OSF.IO/PGCB3.

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
