## [Reviewer comments · BMJ Open]

ARTICLE DETAILS

TITLE (PROVISIONAL)	The acceptability of addressing alcohol consumption as a modifiable risk factor for breast cancer: a mixed method study within breast screening services and symptomatic breast clinics
AUTHORS	Sinclair, Julia; McCann, Mark; Sheldon, Ellena; Gordon, Isabel; Brierley-Jones, Lyn; Copson, Ellen

VERSION 1 - REVIEW

REVIEWER	Emma R Miller Flinders University, Australia
REVIEW RETURNED	27-Nov-2018

GENERAL COMMENTS	This is a nicely written paper on attitudes to alcohol as a modifiable risk factor for breast cancer. This is a very important issue and is of relevance to the journal readership, however there are some fairly major methodological and other issues (some major and some minor) with the study as described below. Specific issues are as follows: 1. Page 3, line 1: Article summary: Strengths and limitations. Doesn't list any strengths or limitations BACKGROUND Overall a good background with some minor grammatical issues: 2. Page 4, from line 34. This is a very long sentence, I suggest rewriting as two sentences.3. Page 5, from line 37. This small paragraph is formed from a single sentence. I suggest rephrasing and connect either to the end of the preceding or beginning of the following paragraph.4. Page 5, from line 45. The first sentence of this paragraph needs restructuring. METHODS:
---

5. This section does not provide sufficient detail overall, especially around the survey development and data analysis.

6. In the first paragraph (page 6), the authors state that the sample sizes were 'agreed as sufficient'. Agreed by whom and on what basis?

7. The authors state that the study was funded on a restricted budget and this is why no sample size calculations were conducted. However, some idea about whether the design is going to be able to answer the research questions is required. Sample size calculations are not a budgetary concern, usually.

8. Also, some statement about how the qualitative sample size was determined is required.

9. Page 6, line 41: what is CRUK? Also, not all international readers would know what BUPA is. These two entities require a brief explanation.

10. A minor issue, but worth rectifying, is that throughout this paper 'data' is referred to as singular rather than plural term.

11. Page 7, first paragraph: the authors describe a literature review that was conducted to inform the questionnaire. What databases and time periods were searched? Was this a systematic search? If so, what were the search terms?

12. Page 7, Quantitative data section: This section needs far more detail about the survey, what was in it and how the questions derived. I am aware that the authors have provided the survey as an additional file but the important details, especially derivation, need to appear in the methods section. It appears that none of the survey were derived from validated instruments or piloted?

13. I have some specific questions in relation to the attached survey, as follows:

a. Question 4: Why do you care about l'egal, marital or same-sex civil partnership status? What about de facto? Or is this covered by 'civil partnership'. If so, how can never married be lumped in with a de facto relationship? (this item doesn't even feature in the analysis).

i. Why do you care about if the person is 'separated but still legally married'? either the person is socially supported in a relationship or not.

b. Question 5: The question asks for 'main occupation' but only lists employment arrangements rather than actual occupations. Also, wouldn't you want to know if they are full time or part time?

c. Section 4: These questions are about health prevention sessions in general. I don't think that the answers would be the same for alcohol prevention sessions - a very loaded topic!

14. Page 7, Qualitative data section: This section also does not have sufficient detail but mentions the attached topic guides.

a. The topic guides state they were developed prior to amendment with information provided by the survey. What was the final schedule?

b. In the interview schedule for opinion leaders and clinic staff, the question is prefaced by (B.1.): "The questionnaire shows that staff

rarely discuss breast cancer risks with patients.” This was not one of the items collected on the attached study survey.

15. From page 8, Analysis section: given the qualitative arm was informed by the survey, quantitative analysis should be described first.

RESULTS:

16. Page 9, reference to Table 1: I don't understand the point of presenting statistical tests comparing the three participant groups. From the table (and intuitively) it is clear that staff differ on just about every parameter listed, for obvious reasons. It is neither valid to pool this information ('totals' in table 1) nor appropriate to subject these widely different categories to a single Chi-square test for each variable.

17. Page 10, from line 25: The authors state there is a “positive correlation with being able to correctly identify the alcohol content of drinks” (despite no evidence of a correlation test being undertaken). An association does not seem at all surprising as it stands to reason that drinkers might read alcohol product labels and non-drinkers give the matter little thought. However, the authors seem to be linking this awareness with propensity to respond to prevention messages. It is important to note that increased knowledge does not equate with willingness for behaviour change.

18. Page 10, line 50: The authors report a result from a regression but do not say what type of regression it was or if there were any cofactors in the model.

19. Page 11, re ‘Potential impact of adding prevention information’ section: According to the provided survey, alcohol prevention was not specifically mentioned as a prevention focus - I suspect that mentioning a focus on alcohol prevention would have greatly influenced responses.

20. From Page 12, Qualitative findings: This section does not seem to reflect a systematic framework of analysis but pure description according to pre-determined headings. There is also a sense of diversion of views etc.

21. Page 13, line 10: Is ‘staff patriarchy’ the correct term here?

DISCUSSION:

22. In the first paragraph, ABI is suddenly introduced and it is not clear how this follows from the findings. Alcohol Brief Interventions are about addressing problematic drinking. This is not relevant to the issue or population studied here.

23. Page 21, Strengths and Limitations: I think you need to add that the survey was not a validated tool. Also that the principle parts of it (what do you think about prevention education and alcohol consumption) were not actually linked in the survey.

[I have attached a file with full formatting in case this is difficult to follow]

REVIEWER	Jane Hughes University of Sheffield
REVIEW RETURNED	10-Dec-2018

GENERAL COMMENTS	Title - I am not sure if the title summarises what the manuscript is about. Is it addressing alcohol consumption as a modifiable risk factor for breast cancer - as stated - or is it something like 'addressing the feasibility/acceptability of incorporating alcohol advice into screening sessions etc' Method - design - ' statement that a minimum size of 100 for each clinical group and 20 participants within the staff group was agreed as a sufficient number'. Would it be possible to state how these numbers were generated - is that based on examining similar study numbers? Limitations - I wonder if another limitation of the intervention being in the screening services clinic might be the fact that screening invitations stop at 70 - even though age is the biggest risk factor in breast cancer - and it might be worth mentioning that this could be an important component of the at risk population who would not be able to benefit from this intervention? Overall - I feel this study is interesting and well thought out and provides important information about an opportunity to intervene in a potentially difficult topic - alcohol consumption and breast cancer.
---

VERSION 1 – AUTHOR RESPONSE

Reviewer 1

Comments to author:

“This is a nicely written paper on attitudes to alcohol as a modifiable risk factor for breast cancer. This is a very important issue and is of relevance to the journal readership, however there are some fairly major methodological and other issues (some major and some minor) with the study as described below”.

We thank this reviewer for their positive comments and for acknowledging the importance of this issue.

1. Page 3, line 1: Article summary: Strengths and limitations.

Doesn't list any strengths or limitations

These have been added

BACKGROUND

Overall a good background with some minor grammatical issues:

2. Page 4, from line 34.

This is a very long sentence, I suggest rewriting as two sentences.

This sentence has now been reworded as follows:

Awareness of alcohol as a risk factor for breast cancer is low [14], as is the ability to estimate the alcohol content of commonly consumed drinks and therefore objectively assess one's own alcohol consumption level [15]. This suggests that many women may be unaware that their level of alcohol consumption may be increasing their risk of breast cancer.

3. Page 5, from line 37.

This small paragraph is formed from a single sentence. I suggest rephrasing and connect either to the end of the preceding or beginning of the following paragraph.

We have moved this sentence to the beginning of the following paragraph.

4. Page 5, from line 45.

The first sentence of this paragraph needs restructuring.

This sentence has now been changed to read as follows:

"This study aimed to assess the acceptability to female patients, and health care staff, of offering an alcohol brief intervention (ABI) in NHS clinical breast services."

METHODS:

5. This section does not provide sufficient detail overall, especially around the survey development and data analysis.

We have clarified this further in response to points (#6, #7, #8, #12, #14 below)

6. In the first paragraph (page 6), the authors state that the sample sizes were 'agreed as sufficient'. Agreed by whom and on what basis?

In addition to adding a sentence to the background section (see #7 below) to clarify the design aims, the wording of this section has been altered and referenced as follows:

Recruitment based on securing a minimum sample size of 100 for each clinical group and 20 staff members was determined to be plausible based on known attendance and staff numbers at Southampton breast services and to be sufficient to generate data of risk factor and alcohol knowledge to be explored in depth within qualitative interviews [21]

7. The authors state that the study was funded on a restricted budget and this is why no sample size calculations were conducted. However, some idea about whether the design is going to be able to answer the research questions is required. Sample size calculations are not a budgetary concern, usually.

One aim of this survey was to generate estimates to be used to determine sample size for a definitive project because there is no published data on variability or effect sizes in relation to alcohol breast cancer awareness or risk factor knowledge. This was part of the protocol for this study (which we have uploaded to the Open Science Foundation depository). A sentence to this effect has now been included in the background section of the manuscript:

These data will be used to inform the sample size for a definitive project as currently there are no published data on variability or effect sizes in relation to alcohol breast cancer awareness or risk factor knowledge.

The design section has also been amended and referenced (see #6 above) which clarifies this point:

8. Also, some statement about how the qualitative sample size was determined is required.

We have clarified this as follows:

Three focus groups consisting of a maximum of eight women each were planned as eight is considered the optimum focus group number when the topic is of importance to participants [22]. Women self-selected and attended the group most convenient for them.

9. Page 6, line 41: what is CRUK? Also, not all international readers would know what BUPA is. These two entities require a brief explanation.

This sentence has been adjusted to read as follows:

The design for this study was developed at an innovation workshop funded jointly by Cancer Research UK and BUPA Foundation (a private health care charity which funds projects and initiatives that make a direct impact on people's health and wellbeing in the UK) which included PPI representation.

10. A minor issue, but worth rectifying, is that throughout this paper 'data' is referred to as singular rather than plural term.

Thank you for spotting this. All references to data have been amended to refer to the plural item.

11. Page 7, first paragraph: the authors describe a literature review that was conducted to inform the questionnaire. What databases and time periods were searched? Was this a systematic search? If so, what were the search terms?

This was not a systematic review but a scoping review of the current literature. We have therefore now rephrased this sentence as follows:

Based on the authors' knowledge of the area and a scoping review of the literature current known modifiable and non-modifiable risk factors for breast cancer were identified and cross-referenced against risk factors reported on public information websites [21-23]

12. Page 7, Quantitative data section: This section needs far more detail about the survey, what was in it and how the questions derived. I am aware that the authors have provided the survey as an additional file but the important details, especially derivation, need to appear in the methods section. It appears that none of the survey were derived from validated instruments or piloted?

We did not find any previous validated measures relating to knowledge of breast cancer risk factors, or knowledge of UK alcohol units in drinks. This was the justification for this preliminary study to inform the use of appropriate measures for intervention development in this context. However we acknowledge that the methods section is currently lacking in detail and this has now been re-written as follows:

As there are currently no psychometrically validated tools to measure knowledge of breast cancer risk factors, or estimation of alcohol content in UK drinks we developed our own questionnaire specifically for this study (see Appendix B). Participant demographic information was collected using standardised UK national statistics demographic categorisation.

13. I have some specific questions in relation to the attached survey, as follows:

a. Question 4: Why do you care about legal, marital or same-sex civil

partnership status? What about de facto? Or is this covered by 'civil partnership'. If so, how can never married be lumped in with a de facto relationship? (this item doesn't even feature in the analysis).

i. Why do you care about if the person is 'separated but still legally married'? either the person is socially supported in a relationship or not.

The categories were based on standardised UK national statistics demographic characteristics as used in the UK national census. These variables did not feature in analysis, but allow comparison of the sample with regional and National demographic characteristics. The origin of these descriptive categories has now been clarified in the text of the manuscript

b. Question 5: The question asks for 'main occupation' but only lists employment arrangements rather than actual occupations. Also, wouldn't you want to know if they are full time or part time?

Please see answer to 13 c) above. Individual occupations are not presented due to small numbers. This information was again use to provide a sample comparison with National statistics, as part of demonstrating the feasibility of this work, rather than as an analytical variable.

c. Section 4: These questions are about health prevention sessions in general.

don't think that the answers would be the same for alcohol prevention sessions

- a very loaded topic!

The introduction to Section 4 in the questionnaire states specifically "We are interested to know your thoughts on the impact of adding specific cancer prevention information to the breast cancer screening process"

We used this phrase as we did not wish to influence participant's responses to other questions within the survey by highlighting alcohol as a risk factor. Alcohol and obesity are the two greatest potentially modifiable risk factors for breast cancer, and would be the target of any specific cancer prevention information. We have now explained this in the methods section. In the discussion section we have explored already the potential ambivalence about discussing alcohol use and its normalisation in society.

14. Page 7, Qualitative data section: This section also does not have sufficient detail but mentions the attached topic guides.

The qualitative data analysis section has been expanded to clarify the method of analysis used which we hope will also address question 20 (below).

a. The topic guides state they were developed prior to amendment with information provided by the survey. What was the final schedule?

To clarify this: The aim of a topic guide in qualitative research is to guide the conversation/ discussion and to ensure that certain areas of interest are covered. They are not meant to be 'questionnaires'. The ethics approval for this study was based on it being a mixed methods study, such that we started with a survey and then explored in greater depth the answers that came out of the survey do understand them better and any further contextual factors that come up during the research, as survey responses are by their nature constrained to the construct imposed upon them by the researcher. The topic guide states at the top "exact nature of these questions will be determined in

part by the responses to the questionnaire". This was the final version of the topic guide approved by the ethics committee.

As can be seen from the second point to be discussed it states in the topic guide "Responses to the questionnaire indicated that x, y, and z are perceived as the most significant risk factors for cancer. To what extent do you agree with these (or not)? Why (or why not)?" The intention is that they are not 'finalised' beyond this level. We have amended Appendix C (see also 14b below) to clarify this

b. In the interview schedule for opinion leaders and clinic staff, the question is prefaced by (B.1.): "The questionnaire shows that staff rarely discuss breast cancer risks with patients." This was not one of the items collected on the attached study survey.

We thank the reviewer for their careful reading as it has identified a mis-transcription from the original guide approved by ethics to the summary form which we thought easier for readers which states:

With the caveat that some of these questions will relate to/ be modified in relation to responses from the questionnaire:

1) The questionnaire shows that KOLs frequently/occasionally/never discuss breast cancer risks with patients?

- Why do you think that is?
- What makes it difficult /easy to do this i.e. what motivates KOLS to engage with this process.

Prompts:

- Is it time?

Organisational/ departmental/professional/personal goals and values? Are there cultural/ organisational/ professional role barriers to doing this easily/ frequently?

We wished to have this question explored with clinic staff, and believed that answers from staff to section 4 of the questionnaire (Q13- Q16) would give us some information (as it did – see qualitative section of the results). We have modified the summary schedule on this, and hope this makes it clearer. If the editors would prefer we could upload the actual guides including potential prompts (rather than the amended summary table), but we felt that this might make things less clear for the reader.

15. From page 8, Analysis section: given the qualitative arm was informed by the survey, quantitative analysis should be described first.

Thank you - the quantitative analysis section has now been placed before the qualitative analysis section.

RESULTS:

16. Page 9, reference to Table 1: I don't understand the point of presenting statistical tests comparing the three participant groups. From the table (and intuitively) it is clear that staff differ on just about every parameter listed, for obvious reasons. It is neither valid to pool this information ('totals' in table 1) nor appropriate to subject these widely different categories to a single Chi-square test for each variable.

Intuitively one could hypothesise differences between the groups; the statistical tests confirm these hypothesised differences, and the tests are statistically appropriate for this study sampling design.

The totals column provides summary overview information for all respondents included in the study; but the sample is analysed in separate subgroups for the reasons the reviewer mentions e.g. the analyses of risk factor awareness excludes staff, because staff knowledge would not be a target for a patient facing intervention.

However we do believe that it is valid, indeed important, to compare the groups, as any cancer prevention initiative is an interaction between the people receiving it, and those responsible for its delivery. The age, BMI, smoking and alcohol consumption status of health professionals, as well as whether they have ever attended screening or a symptomatic clinic is likely to have a significant effect on their attitudes towards delivering cancer prevention advice, and helps triangulate the data found in the qualitative results. Staff members who feel confident about discussing levels of alcohol consumption are more likely to do so than those who see it as outside of their competence to do so. We would request that these tables might remain as they are.

17. Page 10, from line 25: The authors state there is a “positive correlation with being

able to correctly identify the alcohol content of drinks” (despite no evidence of a

correlation test being undertaken). An association does not seem at all surprising as it stands to reason that drinkers might read alcohol product labels and non-drinkers give the matter little thought. However, the authors seem to be linking this awareness with propensity to respond to prevention messages. It is important to note that increased knowledge does not equate with willingness for behaviour change.

We have amended the manuscript to read: “as well as a positive association with being able to correctly identify the alcohol content of drinks” The analysis reports on the association between knowledge of units and knowledge of alcohol as a risk factor for cancer, rather than the association with drinking status. We agree that increased knowledge/ awareness of a risk is not necessarily sufficient to result in behaviour change, but argue it is a necessary prerequisite. We have amended the manuscript as follows:

Overall in both SG and CG there was a significant association between identifying alcohol as a risk factor for breast cancer and personal alcohol consumption (25.2% in those who drink alcohol vs 10.9% in non-drinkers $p=0.031$), as well as a positive association correlation with being able to correctly identify the alcohol content of drinks (of those who got none of the four drink units correct, 86.2% also did not identify alcohol as a risk factor compared to 13.8% who did identify alcohol as a risk factor ($p=0.01$), suggesting that increased awareness about alcohol is associated with the knowledge necessary (if not sufficient) to facilitate behaviour change (see Table 3).

18. Page 10, line 50: The authors report a result from a regression but do not say what

type of regression it was or if there were any cofactors in the model.

The manuscript has been amended to refer to 'unadjusted logistic regression', and regression coefficients prefixed with Odds Ratio (OR) = .

19. Page 11, re 'Potential impact of adding prevention information' section: According to the provided survey, alcohol prevention was not specifically mentioned as a

prevention focus - I suspect that mentioning a focus on alcohol prevention would have greatly influenced responses.

Please see our response to point 13. The aim is not about alcohol prevention, but empowering women to be aware of risks (including alcohol) that they may (or may not) wish to modify, unlike their age, gender or genetics over which they have less control.

20. From Page 12, Qualitative findings: This section does not seem reflect a systematic framework of analysis but pure description according to pre-determined headings. There is also sense of diversion of views etc.

The gold standard, qualitative method thematic analysis was used (also called constant comparison); the intention is to build interpretations based on what participants say and therefore which are grounded in the data. Explanation of what the method is and how it has been used has been given in the analysis section (Page 9) and supported by a reference (Charmaz 2014), to contextualise the process for the reader. Headings of the qualitative findings now describe overarching themes identified in analysis (the previous use of questions as headings was to help orientate the reader, but this has resulted in confusion with analytic method, and so have been removed) . The text describes commonalities as well as divergences in participants' views both between and within groups. This is to give a fuller and richer account of what was reported, and is considered to be a key strength of qualitative approaches.

21. Page 13, line 10: Is 'staff patriarchy' the correct term here?

We have altered this sentence to read:

"Women's expectations of the role of health professionals in providing health education were also apparent here"

DISCUSSION:

22. In the first paragraph, ABI is suddenly introduced and it is not clear how this follows from the findings. Alcohol Brief Interventions are about addressing problematic drinking. This is not relevant to the issue or population studied here.

ABI do address problematic alcohol consumption, but they are most commonly used 'opportunistically' in a range of health care settings (e.g primary care, pharmacies) where people are presenting with a health concern that may (or may not) be related to alcohol. ABI, simply reflect back to a person what they are drinking (based on their responses to a brief screening questionnaire) and make them aware of potential harms. As 'lower risk' alcohol guidelines are being updated (UK and Australia most recently) the carcinogenic risks from alcohol are increasingly acknowledged. As our results show, many women are completely unaware that alcohol is a risk factor for breast cancer (from low levels) and struggle to know how much alcohol is contained in commonly occurring drinks, and so will be unable to know what is their level of drinking. If the editor feels that a more detailed definition of ABI is warranted, we will add this to the paper.

23. Page 21, Strengths and Limitations: I think you need to add that the survey was not a validated tool.

We have added the following statement to the discussion:

Given the lack of research in this area, a survey, utilising a range of response modes, including free text responses was used to explore knowledge of alcohol as a risk factor for breast cancer in the absence of a validated tool.

Also that the principle parts of it (what do you think about prevention education and alcohol consumption) were not actually linked in the survey.

We have clarified in the conclusion that: women participating in this study reacted positively to the suggestion of adding information on cancer prevention, in general, to NHS Breast screening or clinic attendances,

and that

Further research is also required to understand how best to embed a prevention culture, which includes giving clear non-judgemental information about the relative risks of alcohol consumption, within current health systems.

Reviewer: 2

Reviewer Name: Jane Hughes

Institution and Country: University of Sheffield Please state any competing interests or state 'None declared': None declared

Please leave your comments for the authors below:

Title - I am not sure if the title summarises what the manuscript is about. Is it addressing alcohol consumption as a modifiable risk factor for breast cancer - as stated - or is it something like 'addressing the feasibility/acceptability of incorporating alcohol advice into screening sessions etc'

Thank you, we have amended to:

The acceptability of addressing alcohol consumption as a modifiable risk factor for breast cancer: a mixed method study within breast screening services and symptomatic breast clinics

Method - design - 'statement that a minimum size of 100 for each clinical group and 20 participants within the staff group was agreed as a sufficient number'. Would it be possible to state how these numbers were generated - is that based on examining similar study numbers?

Thank you we have amended and referenced this as per comment #6 and #7 to reviewer 1 above

Limitations - I wonder if another limitation of the intervention being in the screening services clinic might be the fact that screening invitations stop at 70 - even though age is the biggest risk factor in breast cancer - and it might be worth mentioning that this could be an important component of the at risk population who would not be able to benefit from this intervention?

We agree that limiting this to the screening service would reduce the exposure of women over the age of 70 to the potential intervention. However in this study (see table 1), 20% of the women who came to symptomatic clinic (the vast majority of whom did not have breast cancer) were over the age of 70, so the over 70's are represented in this study, but we will bear this in mind for the delivery of the intervention if it is limited to screening services.

Overall - I feel this study is interesting and well thought out and provides important information about an opportunity to intervene in a potentially difficult topic - alcohol consumption and breast cancer.

Thank you.

VERSION 2 – REVIEW

REVIEWER	Emma Miller Flinders University, Australia
REVIEW RETURNED	08-Feb-2019

GENERAL COMMENTS	Thanks to the authors for their thoughtful response to my comments and those of the other reviewer. The paper is much
---

	improved from the original submission. I do have some remaining concerns about aspects that I do not feel were adequately addressed by the authors, as follows:  1. Table I accept the authors' response that it is valid to compare some of the characteristics of both those implementing and receiving interventions but I still not believe that it is valid to pool, and subject the differences in groups to statistical analysis, all of those characteristics listed in the table. While comparisons of age and attitudes and perspectives might be interesting, there is little meaning in comparing characteristics such as BMI, employment status, general health, and treatment status in the manner presented. I suggest the authors remove the health workers from this table and find other ways to compare more salient indicators of attitudes to discussion of prevention messages around alcohol. It might be worth just comparing alcohol and smoking status among health workers and patients separately. 2. ABI - ABI is usually a series of linked interventions based on responses to Alcohol screening, usually where there are indications of alcohol use disorder (rather than the sort of drinking likely to increase risk of cancer). The authors need to briefly describe what alcohol screening tool is to be used and what the interventions are. If this is just about asking patients about how much they drink, then providing education around reducing risk in response - this still needs to be described. 3. The authors have not included their completed STROBE checklists.
--	--

REVIEWER	Jane Hughes University of Sheffield England
REVIEW RETURNED	11-Feb-2019

GENERAL COMMENTS	The authors have addressed the points raised in the three reviews and I feel this has an interesting and worthwhile contribution to make to this area of research
---

VERSION 2 – AUTHOR RESPONSE

Reviewer: 2

The authors have addressed the points raised in the reviews and I feel this has an interesting and worthwhile contribution to make to this area of research

Reviewer: 1

Thanks to the authors for their thoughtful response to my comments and those of the other reviewer. The paper is much improved from the original submission. I do have some remaining concerns about aspects that I do not feel were adequately addressed by the authors, as follows:

1. Table I accept the authors' response that it is valid to compare some of the characteristics of both those implementing and receiving interventions but I still not believe that it is valid to pool, and subject the differences in groups to statistical analysis, all of those characteristics listed in the table. While comparisons of age and attitudes and perspectives might be interesting, there is little meaning in

comparing characteristics such as BMI, employment status, general health, and treatment status in the manner presented. I suggest the authors remove the health workers from this table and find other ways to compare more salient indicators of attitudes to discussion of prevention messages around alcohol. It might be worth just comparing alcohol and smoking status among health workers and patients separately.

Table 1 provides a summary of the demographics of all survey participants; staff were an important survey group. We accept that it may not add much meaning to compare the groups in terms of age, employment etc. and so have taken these statistics out of the table as requested. However, we believe that comparing the use of alcohol, and smoking status between staff and women attending breast clinics/ mammograms is valid and important as staff own behaviours have an impact on how comfortable they feel in discussing the same behaviours with patients in their professional role. Therefore we have now included the statistical results relevant to these specific comparisons within the text of the manuscript to support these points.

2. ABI - ABI is usually a series of linked interventions based on responses to Alcohol screening, usually where there are indications of alcohol use disorder (rather than the sort of drinking likely to increase risk of cancer). The authors need to briefly describe what alcohol screening tool is to be used and what the interventions are. If this is just about asking patients about how much they drink, then providing education around reducing risk in response - this still needs to be described.

The aim of this Cancer Research UK funded study was to explore the knowledge and attitudes of staff and attenders at breast health clinics, and explore views "on the impact of adding specific cancer prevention information to the breast cancer screening process "as part of encouraging a cancer prevention culture in breast health settings.

Additionally at focus groups we asked attendees to:

"to discuss their opinion about some of the methods available to try and reduce the number of women who develop breast cancer".

These research questions – and information presented to participants - did not relate to any specific intervention and we did not orient participants to think about a specific ABI protocol. We have highlighted where this is mentioned in the conclusion of the abstract and the opening paragraph of the discussion.

The next phase of this work was to generate the evidence for the logic model of the intervention and then to develop and pilot the intervention prototype that would consist the ABI. This is the subject of a separate project funded by the UK MRC (MRC PHIND (MR/P016960/1), and the results of that study are under analysis at the moment. Details of the development of logic model and the pilot study are beyond the scope of this publication and will be published separately in the future.

We have however now clarified this situation by adding an additional sentence to the discussion of this manuscript as follows:

"Data collected in this study has been used to inform a logic model which will underpin the development of an intervention prototype".

3. The authors have not included their completed STROBE checklists.

This is a mixed methods study, and so the STROBE checklist is only applicable to parts of it. We completed the STROBE online checklist at the beginning of the BMJ Open submission process and updated the article accordingly, Please could the editor now clarify whether this checklist can be made available by BMJ Open to reviewer 1 for their information, or whether we now need to additionally complete a paper version of the STROBE checklist for the benefit of reviewer 1?